# PMSM Torque Ripple Suppression Method Based on SMA-Optimized ILC

**DOI:** 10.3390/s23239317

**Published:** 2023-11-21

**Authors:** Haoyu Li, Yingqing Guo, Qiang Xu

**Affiliations:** College of Mechanical and Electronic Engineering, Nanjing Forestry University, Nanjing 210037, China; leehaoyu@njfu.edu.cn (H.L.);

**Keywords:** SMA, iterative learning, PMSM, torque ripple suppression

## Abstract

Periodic torque ripple often occurs in permanent magnet synchronous motors due to cogging torque and flux harmonic distortion, leading to motor speed fluctuations and further causing mechanical vibration and noise, which seriously affects the performance of the motor vector control system. In response to the above problems, a PMSM torque ripple suppression method based on SMA-optimized ILC is proposed, which does not rely on prior knowledge of the system and motor parameters. That is, an SMA is used to determine the optimal values of the key parameters of the ILC in the target motor control system, and then the real-time torque deviation value calculated by iterative learning is compensated to the system control current set end. By reducing the influence of higher harmonics in the control current, the torque ripple is suppressed. Research results show that this method has high efficiency and accuracy in parameter optimization, further improving the ILC performance, effectively reducing the impact of higher harmonics, and suppressing the torque ripple amplitude.

## 1. Introduction

The permanent magnet synchronous motor (PMSM) has superior performance and advantages in terms of efficiency, power density, torque-to-inertia ratio, and reliability. However, factors such as cogging torque and harmonic flux in the motor can cause distortion in the control current, resulting in periodic fluctuations in the motor output torque. This fluctuation is known as torque ripple in the motor. This phenomenon can cause mechanical vibration, speed fluctuations [1], and acoustic noise at high speeds in the motor [2], which has a significant impact on the high-precision positioning in the speed control and position control systems at low speeds, greatly reducing the control performance of the servo system. Therefore, motor torque ripple suppression is extremely important in high-precision servo control systems.

The PMSM is a multi-variable, strong-coupled nonlinear system, whose periodic torque ripple involves various elements of the motor, such as harmonics, cogging torque, current identification error, phase imbalance, etc. [3]. Currently, research on the PMSM periodic torque ripple suppression can be divided into two routes: one is based on the motor body, which eliminates vibration through the optimization of the mechanical structure of the motor body, such as using closed slots, but this will lead to large leakage flux, reduced electromotive force of the armature winding, and increased inductive resistance. These shortcomings can be compensated for by increasing the excitation magnet-motive force and using removable teeth [4]. In addition to this, there are methods to optimize the motor structure by improving the winding distribution [5], stator skew slot, or rotor skew pole [6], optimizing magnetic pole shape, etc. [7]. However, motors with different functions have different body structures; therefore, the strategy of optimizing the motor body to eliminate vibration has poor versatility. The other is based on the control strategy, which controls the electromagnetic parameters of the motor and compensates the motor control current or voltage without changing the structure of the motor body; thereby, suppressing the torque ripple [8].

With the widespread application of the PMSM in industrial production in recent years, torque ripple suppression methods based on control strategies have been intensively studied. Chen et al. [9] suppressed the torque ripple caused by flux harmonics with the injection of a 5th and a 7th high-frequency harmonic current into the control signal. However, the calculation formula is complex, and it is difficult to set the control parameters. Qu et al. [10] derived the relationship between the amplitude of velocity harmonics and harmonic current based on the harmonic injection method, and proposed a new velocity harmonic controller considering the copper loss, which reduces the impact of velocity harmonic phase. Doo-Il et al. [11] use the extended Kalman filter to predict the distorted output voltage and compensate for the dead-time effect of the inverter to improve control stability. However, this method has a limited effect on suppressing torque ripple caused by other causes. Zhang et al. [12] implemented a zero steady-state error tracking control for the sinusoidal signal at the resonant frequency and connected the resonant controller in parallel with the current loop PI controller to improve the current waveform of the stator and control the torque ripple. Gao et al. [13] proposed an online compensation control method using a hybrid torque distribution function to control current fluctuations and optimize torque distribution to reduce torque ripple. Song et al. [14] defined the double rotor position angle to apply two different operating modes, thereby performing different torque lag control schemes, based on the ampere torque characteristics of the motor. However, this method is complex and difficult to apply in industrial production. Li B. et al. [15] proposed a torque ripple suppression strategy of an ILC (iterative learning control) algorithm. By utilizing the obvious periodic characteristics of torque ripple, an online iterative approach is used to obtain an accurate torque tracking trajectory, and the deviation signal is compensated to the q-axis current to achieve torque ripple suppression. On this basis, Ji P. et al. [16] used the improved particle swarm optimization (PSO) to optimize and tune the key PID parameters in the ILC, thereby improving the torque ripple suppression effect. However, in practical applications, when the PSO is optimized, the convergence speed is slow, and it is easy to fall into the local optimum [17].

The slime mold algorithm (SMA) is a new meta-heuristic algorithm proposed by Li et al. [18] in 2020, which mainly uses mathematical models to simulate the behavior and morphological changes of slime molds in the foraging process in nature. The weight coefficient of the SMA simulates the positive and negative feedback generated by the biological oscillator of the slime mold when encountering food of different concentrations: when high-quality food is found, the slime mold will approach quickly; when the food concentration is low, the slime mold will move slowly, thereby approaching the best food source with greater efficiency. SMA code has a simple structure, strong scalability, and excellent performance in function optimization and engineering design, with the advantages of fast response and strong anti-interference in parameter self-tuning [19].

Therefore, this paper proposes a torque ripple suppression method with a forgetting factor for an open-closed-loop iterative learning control to address the periodic torque ripple generated during the operation of a PMSM. The electromagnetic torque output by the system is used as a feedback signal to track the output current of the speed loop. The difference between the reference signal and the output signal is compensated in real time to the q-axis current reference terminal through an iterative learning control algorithm, and the iteration is repeated continuously to reduce the error. The chaos-mapped SMA is used to optimize the PID control parameters and forgetting factors in the iterative learning controller, improving the tracking accuracy and convergence speed of the torque error. This method does not rely on the system’s prior knowledge and motor parameters, and can adapt to the target motor, reducing manual debugging of parameters. It improves the optimization speed, avoids local optima, achieves better suppression effects, and further improves the torque ripple suppression effect. Simulation experiments were performed in the MATLAB/Simulink platform. The reduction in the number of iterations required for adaptive iteration obtained from simulation, as well as the reduction in torque ripple coefficient and torque ripple, all indicate that this method can further improve the performance of iterative learning control and better suppress the amplitude of torque ripple.

Therefore, this paper proposes a torque ripple suppression method of an open-closed loop iterative learning control with forgetting factor for the periodic torque ripple generated when a PMSM is working, and optimizes and tunes the control parameters in the ILC using the SMA after chaos mapping to improve the optimal-searching speed and avoid local optimum, thereby further improving the torque ripple suppression effect. Relevant simulation experiments were conducted on the Simulink platform, and the simulation results showed that this method can further improve the performance of iterative learning control and suppress torque ripple amplitude.

## 2. Mathematical Model of Motor

In the vector control system, neglecting the saturation of the magnetic circuit and core loss, as well as the effect of leakage flux and eddy current effect, the voltage equation of a PMSM in the synchronous rotating coordinate system is:(1)$$\left\{\begin{array}{c} u_{d}=R_{S}i_{d}+\frac{d\psi_{d}}{dt}+\omega_{e}\psi_{q}\\ u_{q}=R_{S}i_{q}+\frac{d\psi_{q}}{dt}+\omega_{e}\psi_{d} \end{array}\right.$$

In the equation, $R_{S}$ represents the resistance of the stator winding; $\omega_{e}$ represents the mechanical angular velocity; $u_{d}$ and $u_{q}\;$stand for the voltage of d-axis and q-axis, respectively; $i_{d}$ and $i_{q}$ stand for the currents of d-axis and q-axis, respectively; while $\psi_{d}$ and $\psi_{q}$ represent the stator flux linkage of d-axis and q-axis, respectively. The equation is:(2)$$\left\{\begin{array}{c} \psi_{d}=L_{d}i_{d}+\psi_{f}\\ \psi_{q}=L_{q}i_{q} \end{array}\right.$$

Among them, $L_{d}$ and $L_{q}$ stand for the inductances of d-axis and q-axis, respectively and $\psi_{f}$ represents the rotor flux linkage.

If further assumed that the PMSM used is a surface-mounted motor without salient poles and $L_{d}$ and $L_{q}$ are equal, then the electromagnetic torque equation of the motor is:(3)$$T_{e}=\frac{3}{2}p(\psi_{d}i_{q}-\psi_{q}i_{d})=\frac{3}{2}p\psi_{f}i_{q}=K_{t}i_{q}$$

In Equation (3), *p* represents the number of pole pairs of the motor and $K_{t}$ represents the torque coefficient.

## 3. Factor Analysis of Torque Ripple

The main causes of torque ripple include cogging torque and flux harmonic distortion; among which, cogging torque is the circumferential torque generated by the interaction between the rotor permanent magnet magnetic field and the cogging of the stator core, achieving suppression through the optimized design of the motor body [20], while magnetic flux harmonic distortion is the main source of periodic torque ripple, and is also the primary target of the control strategy to suppress torque ripple. The permanent magnet excitation magnetic field and the stator winding cannot achieve a completely sinusoidal distribution. As a result, the air-gap flux density distribution cannot achieve a completely sinusoidal waveform, which leads to harmonics in the flux linkage in the permanent magnet and stator current appearing in the 3rd, 5th, 7th, 11th harmonic, etc. [21]. As there is no 3rd harmonic in the Y-shaped connected stator winding, the 5th and 7th harmonics have the greatest impact on the system. The mathematical model of harmonics in the PMSM stationary three-phase coordinates can be expressed as:(4)$$\left\{\begin{array}{c} u_{a}=u_{1}\cos(\theta_{e}+\varphi_{1})+u_{5}\cos(-5\theta_{e}+\varphi_{5})+u_{7}\cos(7\theta_{e}+\varphi_{7})+\cdots \\ u_{b}=u_{1}\cos(\theta_{e}+\varphi_{1}-\frac{2}{3}\pi )+u_{5}\cos(-5\theta_{e}+\varphi_{5}-\frac{2}{3}\pi )+u_{7}\cos(7\theta_{e}+\varphi_{7}-\frac{2}{3}\pi )+\cdots \\ u_{c}=u_{1}\cos(\theta_{e}+\varphi_{1}+\frac{2}{3}\pi )+u_{5}\cos(-5\theta_{e}+\varphi_{5}+\frac{2}{3}\pi )+u_{7}\cos(7\theta_{e}+\varphi_{7}+\frac{2}{3}\pi )+\cdots \end{array}\right.$$

Among them, $u_{a}$, $u_{b}$, and $u_{c}$ stand for three-phase voltages; $\theta_{e}$ represents the electrical angle of the motor; $u_{1}$ and $\varphi_{1}$ stand for the amplitude and initial phase of the fundamental voltage, respectively; $u_{5}$ and $\varphi_{5}$ stand for the amplitude and initial phase of the 5th harmonic voltage, respectively; and $u_{7}$ and $\varphi_{7}$ stand for the amplitude and initial phase of the 7th harmonic voltage, respectively. Under the action of the magnetic field of the permanent magnet, the 6th and 12th harmonic torque equations generated by the 5th and 7th harmonic voltages are:(5)$$\begin{array}{cc} T_{e} & =\frac{3}{2}pi_{q}\psi_{f}(\theta_{e})\\ & {=T}_{0}+T_{6}\cos6\theta_{e}+T_{12}\cos12\theta_{e}+\cdots \\ & {=T}_{0}+T_{ripple} \end{array}$$

In the equation, $T_{0}$, $T_{6}$, and$\;T_{12\;}$represent the amplitude of DC torque, 6th and 12th harmonic torque, respectively; $T_{\mathrm{ripple}}$ is the total harmonic torque. According to Equation (5), it can be seen that the electromagnetic torque of the motor is mainly composed of the fundamental wave component and the 6th and 12th harmonic components. The ultimate control goal is to suppress the periodic torque ripple caused by higher harmonics.

## 4. Iterative Learning Suppression Algorithm of Torque Ripple

An iterative learning control (ILC) is a control method for trajectory tracking systems that perform repeated movements [22], which generates the desired input by applying the previous control information multiple times and continuously iterates the previous control information so that the deviation signal tends to zero within a limited time. For repeatable periodic motion or periodic interference, an ILC can achieve better control effects [23].

In this paper, the author designs a PID iterative learning controller for the speed loop of the PMSM vector control system, which uses the electromagnetic torque output by the system as a feedback signal, tracks the output current of the speed loop, and compensates the q-axis current set end in real time with the difference between the reference signal and the output signal through the ILC algorithm, repeating the iteration to reduce the error. Through this method, the controller can adjust the current in real time to control the output torque of the motor, achieving the torque ripple suppression effect.

### 4.1. Mathematical Model of ILC

The ILC iteratively calculates the error $e_{k}\left(t\right)$ between the desired output $y_{d}\left(t\right)$ and the actual system output $y_{k}\left(t\right)$, in a dynamic repeating system of duration $t\in \left[0,T\right]$:(6)$$\left\{\begin{array}{l} x(t)=f(t,x(t))+Bu(t)+E\\ y(t)=Cx(t) \end{array}\right.$$

$x\left(t\right)$represents the system state variable, $u\left(t\right)$ represents the system input variable, and $y\left(t\right)$ represents the system output variable. The output-controlled quantity of the ILC $\;u_{k}\left(t\right)$ can be expressed as:(7)$$u_{k}(t)=L(u_{k-1}(t),e_{k}(t))$$

$e_{k}\left(t\right)$ represents the error between the expected output value and the actual output value, $u_{k-1}\left(t\right)$ represents the controlled quantity of the previous period, and L is the control rate of iterative learning. Commonly used ILC laws include P type and PI type, PID type, etc., and it can be divided into open loop, closed loop, open-closed loop, etc. according to the feedback. The difference between open-loop iterative learning and closed-loop iterative learning is whether to use the signal stored in the previous moment or the current moment signal. To enhance the robustness of an ILC and accelerate the convergence speed of the algorithm, the forgetting factor α can be introduced, which represents the proportion of the controlled quantity of the previous cycle to the controlled quantity of the current cycle. Therefore, the PID open-closed loop ILC law with the forgetting factor in this paper can be expressed as:(8)$$u_{k}(t)=(1-\alpha )u_{k-1}(t)+k_{p}e_{k-1}(t)+k_{d}\frac{de_{k-1}(t)}{dt}+k_{i}\int_{0}^{t}e_{k-1}(\tau )d\tau$$

The value of the forgetting factor *α* can be between 0 and 1. When the value of *α* is larger, the error of the system output decreases as the value increases. However, when the value of *α* approaches 1, the forgetting factor gradually fails. Therefore, the value of the forgetting factor needs to be determined based on actual engineering conditions to consider the convergence speed and robustness of iterative learning.

### 4.2. ILC Design

An open-closed loop PID type ILC algorithm with a forgetting factor is used to suppress the periodic torque ripple of the PMSM. From Equation (3), it can be seen that the electromagnetic torque of the motor is proportional to the *q*-axis current $i_{q}$ under ideal circumstances, and the expected electromagnetic torque value $T_{e}$ is used as $T_{m}^{*}\;$ the expected output value of the ILC and the torque signal feedback value $T_{m}$ is used as the actual output value. The torque current compensation value $i_{q}^{com}$ can be obtained through an ILC with the error between the two. Therefore, the mathematical description of the ILC law in the motor control system is shown in Equation (9), and the ILC block diagram is shown in Figure 1.
(9)$$i_{q,k}^{com}(t)=(1-\alpha )i_{q,k-1}^{com}(t)+k_{p}e_{k-1}(t)+k_{d}\frac{de_{k}(t)}{dt}+k_{i}\int_{0}^{t}e_{k}(\tau )d\tau$$
(10)$$e_{k}(t)=T_{m}^{*}-T_{m}$$

Among them, $i_{q,k}^{com}\left(t\right)$ and $i_{q,k-1}^{com}\left(t\right)\;$ stand for the torque current compensation values of the current cycle and the previous cycle, respectively; $e_{k}\left(t\right)$ and $e_{k-1}\left(t\right)$ stand for the torque error signals of the current cycle and the previous cycle, respectively; $k_{p}$, $k_{i}$, and $k_{d}$ are the PID gains of the ILC; and α is the forgetting factor.

## 5. ILC Parameter Optimization with SMA

### 5.1. SMA

An SMA (slime mold algorithm) is a mathematical model that simulates the behavior and morphological changes of slime molds during their foraging process in nature. When the slime mold approaches a food source, the biological oscillator generates a propagation wave through the veins, increasing the cytoplasmic flow [24]. The higher the food concentration, the stronger the propagation wave generated by the biological oscillator, and the faster the cytoplasmic flow. An SMA simulates the above-mentioned behavior through a mathematical model, which can be divided into three stages: the food-approaching stage, surrounding food stage, and grabbing food stage. The above process is detailed in Equation (11):(11)$$X(n+1)=\left\{\begin{array}{cc} rand\times (ub-lb)+lb\;\; & r<z\\ X_{b}(n)+vb(W\times X_{a}(n)-X_{b}(n))\; & z<r<j\\ vc\times X(n)\;\; & r\geq j \end{array}\right.$$
(12)j=tanh(|S(i)−DF|)(i=1,2,…,M)
(13)a=arctanh(1−(n/N))

Among them, $X\left(n+1\right)$ and $X\left(n\right)$ are the positions of the slime mold in the n+1th and nth iterations, respectively; $X_{b}\left(n\right)$ represents the position with the highest food concentration at the nth iteration (the best position); $X_{b}\left(n\right)$ and $X_{b}\left(n\right)$ represent two slime mold individuals randomly selected in the nth iteration; vb is a random number in the range [−a, a]; n is the current iterations; N is the maximum iterations; vc is a random number linearly decreasing from 1 to 0; r is a random number between 0 and 1; $S\left(i\right)$ represents the fitness value of the ith slime mold individual; and DF represents the optimal fitness value in all iterations, *M* represents the population size of slime mold, and *W* represents the quality coefficient of slime mold, and its equation is:(14)$$W(SortIndex(i))=\left\{\begin{array}{cc} 1+r\times \log(\frac{bF-S(i)}{bF-wF}+1) & \;condition\\ 1-r\times \log(\frac{bF-S(i)}{bF-wF}+1) & \;others \end{array}\right.$$
(15)SortIndex=sort(S)

Among them, condition represents the individual whose fitness value ranks in the top half of the group; bF and wF represent the optimal fitness value and the worst fitness value in the current iteration, respectively; and SortIndex represents the sorted fitness value sequence. The algorithm updates individual positions based on changes in the optimal position $X_{b}$ and fine-tuning of vb, vc, and W. The function of r is to form a search vector at any angle, that is, to search the solution space in any direction, thereby increasing the possibility of finding the optimal solution and better simulating the circular and fan-shaped structural motion of slime molds approaching food. $W\left(SortIndex\left(i\right)\right)$ simulates the positive and negative feedback process between slime mold weight and food concentration (fitness value) and log is used to slow down the change rate of the value and stabilize the change in contraction frequency. This equation simulates the process of slime molds adjusting their positions according to food concentration; the higher the food concentration, the greater the weight of slime molds near the area. If the food concentration is low, slime molds will turn to search other areas [25].

The search range of the SMA can be further expanded by using logistic chaotic mapping to initialize the population instead of random values. With a simple structure, logistic chaotic mapping can show balance in the phase space through a reasonable control parameters selection. By using logistic mapping during population initialization, the search range can be effectively expanded, and the efficiency of subsequent algorithms can be improved. Its iteration expression is:(16)$$X_{i+1}=rX_{i}(1-X_{i})i=0,1,2,\ldots$$

$X_{i}$ and $X_{i+1}$ are the ith and i+1th chaotic variables, respectively, $X_{i}\in \left[0,\;1\right]$; and r is the control parameter of chaotic mapping, $r\in \left[0,\;4\right]$, which c determines the behavior and properties of the mapping. When r takes a value of 4 and $X_{0}$ takes a random value within [0, 1], pseudo-random values distributed relatively evenly within the range of [0, 1] can be generated.

### 5.2. ILC Parameter Optimization by SMA

This paper selects four key parameters of iterative learning, $k_{p}$, $k_{i}$, $k_{d}$ and forgetting factor α, as the optimization targets of the SMA to obtain better torque ripple suppression effect. The system block diagram is shown in Figure 2. The green dotted box in the figure is the torque ripple suppression module, and the blue dotted box is the SMA-optimized parameter module. When the motor is running, the system collects the error signal $e_{k}\left(t\right)$ between the tracking variable $T_{m}^{*}$ and the torque feedback value $T_{m}$ in real time. The ILC after parameter optimization performed through the SMA calculates the compensation value $i_{q,k}^{com}\left(t\right)$, and compensates it to the set end of the *q*-axis current.

A periodic trigger is incorporated to this process, with the rotor rotating once as a period, and the rotor position signal collected by the system is used to replace the time signal to control the operation of the ILC, thus effectively preventing the deviation of the compensation position corresponding to the current compensation value.

In slime mold algorithms, fitness fis usually used as an evaluation index to judge the quality of individual positions and determine the optimal individual position. In the PMSM vector control system, the system error $e_{k}\left(t\right)$ is first selected as the primary performance index. At the same time, in order to prevent excessive fluctuations in input energy, the operating variance of the compensated q-axis current $i_{q,var}^{*}$ is the secondary influencing factor included in the fitness function. The specific expression of the fitness function is:(17)$$f=\int_{0}^{\infty }(k_{e}\cdot \left|e_{k}(t)\right|\cdot t+k_{u}\cdot i_{q,\mathrm{var}}^{*})dt$$

In this equation, $k_{e}$ and $k_{u}$ are weight coefficients, and $k_{e}>k_{u}.$ The value of the fitness function f  reflects the quality of the slime mold individual position. The smaller the value, the better the control effect of the corresponding individual position on the surface. The individual position corresponding to the minimum value of f is the current optimal solution.

### 5.3. Steps of Algorithm Optimization

The specific steps for an ILC parameter optimization based on the SMA are as follows.

Step 1: Initialization. Use logistic chaotic mapping to initialize the slime mold population positions in the four dimensions of $k_{p}$, $k_{i}$, $k_{d}$, and α, given the maximum iterations *N* and the population size *M*.

Step 2: Assign the positions of $k_{p}$, $k_{i}$, $k_{d}$, and α to the ILC, then run the system to calculate and evaluate the fitness f of the current position according to Equation (17).

Step 3: Update the current optimal individual and its fitness value.

Step 4: Calculate the slime mold mass coefficient W according to Equation (15).

Step 5: For each slime mold individual, update the parameters vb, vc, and p according to Equations (13) and (14), and then update the slime mold individual position according to Equation (12).

Step 6: If the algorithm reaches the set maximum iterations, output the optimal PID control parameters (that is, the optimal position of the SMA); otherwise, return to Step 2.

Step 7: The algorithm ends.

## 6. Simulation Experiment

In Matlab, the function m is used to write the chaotic initialization SMA, and a PMSM double closed-loop vector control (foc) model based on iterative learning compensation torque ripple suppression is built in Simulink to control the four parameters in the ILC for iterative optimization, whose value ranges are $k_{p}$∈ [0, 10], $k_{i}$∈ [0, 10], $k_{d}$ ∈[0, 1], α ∈ [0, 1], with the population size *M* = 24, dimension d = 4, and maximum iterations T = 50. The configuration parameters of PMSM in Simulink are shown in Table 1:

The target parameters are optimized through the SMA and compared with the optimization results of the PSO, to obtain the optimal parameters shown in Table 2, and the fitness function convergence curve shown in Figure 3.

In Figure 3, the convergence speed of the SMA is faster than the PSO. The convergence interval is found in the third iteration, and it jumps out of the local optimum and finds the global optimal fitness value in the 19th iteration, while the PSO falls into a local optimum in the late iteration and has poor global search capabilities.

To further verify the effectiveness of this method, the data of torque suppression effect after ILC-SMA optimization is compared with the data from the system without a suppressor, and the data after being suppressed by the ILC and ILC-PSO methods:

The torque ripple coefficient $K_{TRF}$ is used to evaluate the torque ripple size:(18)$$K_{TRF}=\frac{T_{\max}-T_{\min}}{T_{ave}}\times 100\%$$

In this equation, $T_{\max}$ and $T_{\min}$ represent the maximum and minimum values of torque ripple, respectively, and $T_{ave}$ is the average torque. The torque ripple coefficient and standard deviation of the original system and the three suppression methods are compared:

Combining the simulation results of several suppression methods in Figure 4 and Table 3, it can be seen that when the PMSM is at a constant speed and the load torque is stable, the torque ripple amplitude value output by the open-closed loop ILC-SMA method proposed in this paper is smaller than that of other methods, with standard deviation of torque ripple reduced to 0.0776 and torque ripple coefficient reduced to 3.47%. Compared with the PSO-ILC method, it is reduced by 19.1% and 26.3%, respectively, significantly reducing the intensity of torque ripple near a given load torque, which proves that the torque ripple suppression effect is better than that of the other two algorithms.

To further verify the above conclusion, the fast Fourier transform (FFT) is performed on the torque ripple in the above figure, and the following harmonic analysis comparison chart is obtained.

It can be seen from the harmonic analysis results in Figure 5 and Table 4 that both the ILC method and the PSO-ILC method can effectively reduce the higher harmonic components in the PMSM torque ripple. Compared with the PSO-ILC method, the SMA-ILC method proposed in this paper reduces the 6th harmonic component from 239.736% to 195.551% and the 12th harmonic component from 116.683% to 96.245%, which are reduced by 18.43% and 17.52%, respectively. It proves that this method has a good effect in suppressing the harmonic component of the output torque and can effectively suppress the periodic torque ripple of the PMSM.

## 7. Conclusions

In summary, this article discusses the suppression of torque ripple in permanent magnet synchronous motors and proposes a torque ripple suppression strategy based on SMA optimization iterative learning parameters. In traditional dual closed-loop FOC control systems, the iterative learning deviation value is used to compensate for the speed loop q-axis torque current to reduce periodic torque ripple.

The simulation results show that the SMA-ILC method has a better performance than the PSO-ILC method in terms of torque ripple suppression. Compared with the PSO-ILC method, the standard deviation and torque ripple coefficient of the SMA-ILC method are reduced by 19.1% and 26.3%, respectively, and the high-order harmonics after FFT analysis are reduced by 18.43% and 17.52%, respectively. The results show that the SMA-ILC method has a more effective torque ripple suppression effect for permanent magnet synchronous motors. This scheme can be used in industries with high requirements for low-speed stability of motors, such as the sports equipment industry. It can be used in treadmills to reduce speed fluctuations at low speeds and reduce the vibration felt by the foot. The SMA-ILC algorithm improves efficiency, but does not solve the dependence of this type of control method on torque sensors or torque observers. In future research, deep learning technology can be introduced to consider the non-electric factors affecting torque ripple, further improving the accuracy of torque ripple.

## Figures and Tables

**Figure 1 sensors-23-09317-f001:**
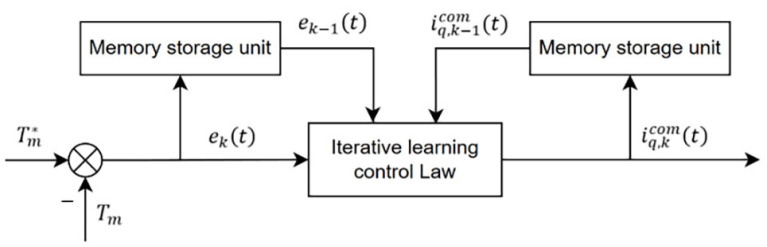
Torque current ILC block diagram.

**Figure 2 sensors-23-09317-f002:**
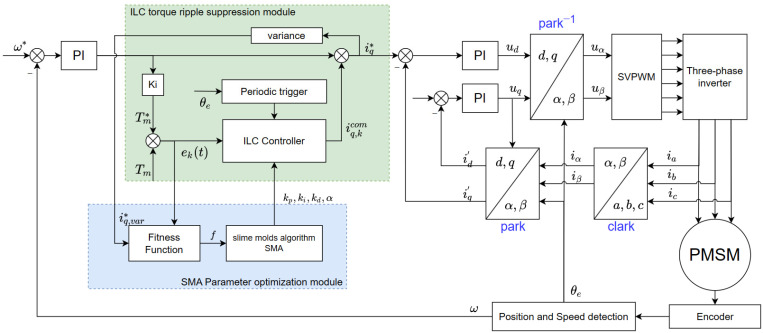
PMSM vector control system based on SMA optimized Iterative learning control.

**Figure 3 sensors-23-09317-f003:**
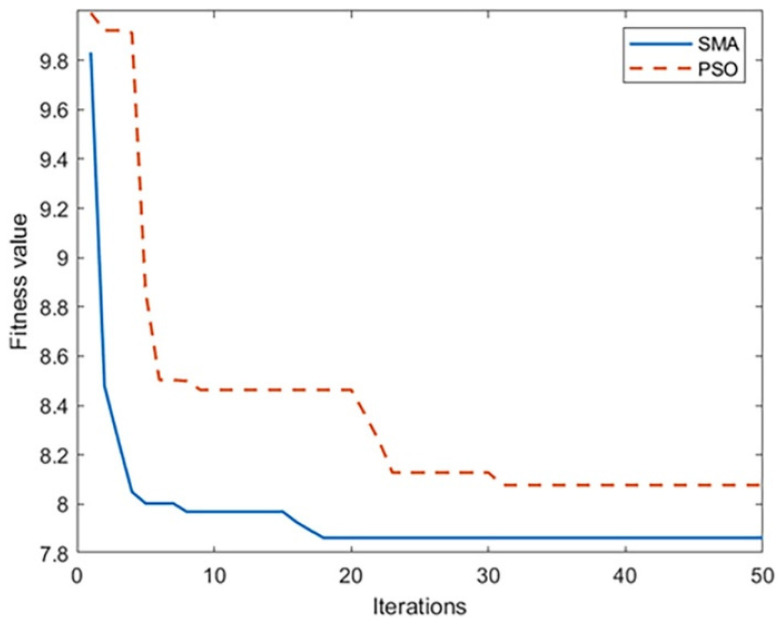
Comparison of fitness function convergence curve.

**Figure 4 sensors-23-09317-f004:**
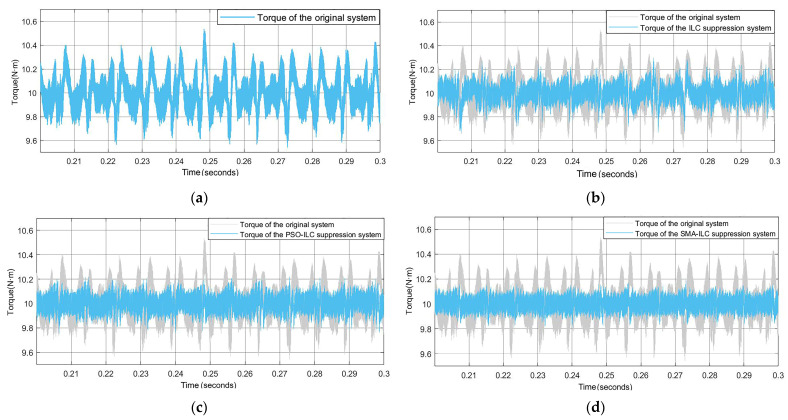
Comparison of torque ripple suppression effects (**a**) system without suppressor; (**b**) ILC suppressing torque ripple; (**c**) PSO-ILC suppressing torque ripple; (**d**) SMA-ILC suppressing torque ripple.

**Figure 5 sensors-23-09317-f005:**
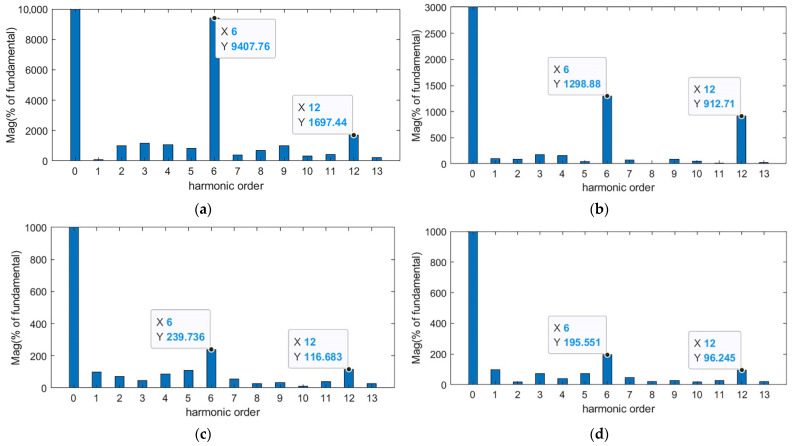
Comparison of FFT harmonic analysis (**a**) system without suppressor FFT; (**b**) ILC suppressing torque ripple FFT; (**c**) PSO-ILC suppressing torque ripple FFT; (**d**) SMA-ILC suppressing torque ripple FFT.

**Table 1 sensors-23-09317-t001:** Control system simulation parameters.

Parameter	Value
Target speed	300 r/min
Load torque	10 N·m
Pole-pairs p	4
Stator resistance $R_{s}$	0.958 Ω
Lss inductance L	$9.48\times 10^{-3}\;$
Permanent magnet flux linkage $\psi_{f}$	0.1872 $\;W_{b}$
Rotational inertia J	0.003 $\mathrm{kg}.m^{2}$
Type of connection	star connection

**Table 2 sensors-23-09317-t002:** Parameter optimization results.

Algorithm	$k_{p}$	$k_{i}$	$k_{d}$	α	f
PSO	0.3440	0.9911	0.0283	0.8434	8.2773
SMA	0.1419	2.0334	0.0332	0.9215	7.8633

**Table 3 sensors-23-09317-t003:** Comparison of torque ripple suppression effects.

Control Algorithm	Torque Ripple Standard Deviation	Torque Ripple Coefficient
System without suppressor	0.1725	9.97%
ILC suppression	0.1053	6.40%
PSO-ILC suppression	0.0959	4.71%
SMA-ILC suppression	0.0776	3.47%

**Table 4 sensors-23-09317-t004:** Comparison of FFT harmonic analysis.

Control Algorithm	6th Harmonic Wave	12th Harmonic Wave
System without suppressor	9407.76%	1697.44%
ILC	1298.88%	912.71%
PSO-ILC	239.736%	116.683%
SMA-ILC	195.551	96.245%

## Data Availability

Data are contained within the article.

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
