# Peer review of "PMSM Torque Ripple Suppression Method Based on SMA-Optimized ILC"

_sensors, 2023, doi:10.3390/s23239317_

Round 1
Reviewer 1 Report
Comments and Suggestions for Authors
1. What is torque ripple in the context of Permanent Magnet Synchronous Motors, and why is it a significant issue in motor control systems?
2. How do cogging torque and flux harmonic distortion contribute to the occurrence of periodic torque ripple in PMSMs?
3. What are the negative effects of torque ripple on motor performance, including motor speed fluctuations, mechanical vibration, and noise?
4. Explain the key concept of using Iterative Learning Control (ILC) to mitigate torque ripple in PMSMs?
5. How does the proposed method use Sequential Monte Carlo Simulated Annealing (SMA) to optimize the parameters of the ILC, and what is the significance of not requiring prior knowledge of the system and motor parameters?
6. What are the critical parameters of the ILC that SMA optimizes, and how does this optimization contribute to torque ripple reduction?
7. Describe the process of compensating the real-time torque deviation value to the system control current set end and how this reduces the influence of higher harmonics in the control current?
8. What is the evidence or data from research results that support the method's efficiency, accuracy in parameter optimization, and its ability to reduce torque ripple amplitude effectively?
9. How does the proposed method compare to other approaches for torque ripple suppression in PMSMs, and what are its advantages or unique features?
10. What potential applications or industries could benefit from this method, and what are the implications for improving motor control systems and performance?
11. Please add the experimental validation of the proposed scheme of control structure.
12. Kindly check the caption of Table 2.
13. Cite all the equations properly.
14. Recent articles can be referred and old references can be removed.
15. Discuss the results with more literature support and identify the background physics behind the control algorithm. How does it perform better?
Comments on the Quality of English LanguageModerate editing of the English language required
Reviewer 2 Report
Comments and Suggestions for Authors
There is another design method for reducing tooth harmonic ripple: closed slots. The disadvantages of this method are large leakage fluxes, reduced EMF and increased inductive resistance of the armature winding, and the technological complexity of winding installation. However, these shortcomings are compensated by an increase in excitation MMF and the use of removable tooth: doi:10.1088/1742-6596/1559/1/012146.
-59 -60; -231 -233 – copy-paste duplicates.
-140 how is the mechanical torque for a feedback loop measured? The mandatory presence of an additional torque sensor imposes significant restrictions on the areas of practical application of this method.
The example shows a high-torque, low-speed electric motor. Is this a consequence of restrictions on the maximum rotation speed due to the insufficient speed of the calculation and control system, or the low sensitivity of the torque sensor?
In Table 1 it is necessary to indicate the type of connection of the armature windings in star or triangle. The presence of the 3rd harmonic in figure 5 contradicts statement -114.
The fonts on the graphs in Figure 4 are small. In the electronic version you can zoom in, but when printed, they are unreadable.
It is possible that the non-traditional values on the horizontal axis of the graphs in Figure 5 are shifted. They start with the zero harmonic, there are even ones. Before the FFT result, it is advisable to show the original periodic function and its period. And indicate the number of armature tooth per pole division.
Round 2
Reviewer 1 Report
Comments and Suggestions for Authors
Congrats to the authors.
Comments on the Quality of English LanguageMinor editing of the English language required